# Effect of Nano-Sized Poly(Butyl Acrylate) Layer Grafted from Graphene Oxide Sheets on the Compatibility and Beta-Phase Development of Poly(Vinylidene Fluoride) and Their Vibration Sensing Performance

**DOI:** 10.3390/ijms23105777

**Published:** 2022-05-21

**Authors:** Miroslav Mrlik, Markéta Ilčíková, Josef Osička, Erika Kutálková

**Affiliations:** 1Centre of Polymer Systems, Tomas Bata University in Zlin, 760 01 Zlin, Czech Republic; osicka@utb.cz (J.O.); ekutalkova@utb.cz (E.K.); 2Polymer Institute, Slovak Academy of Sciences, Dubravska cesta 9, 845 45 Bratislava, Slovakia; 3Department of Physics and Materials Engineering, Faculty of Technology, Tomas Bata University in Zlin, 760 01 Zlin, Czech Republic

**Keywords:** SI-ATRP, graphene oxide, poly(vinylidene fluoride), dielectric properties, compatibility, vibration sensing, d_33_

## Abstract

In this work, graphene oxide (GO) particles were modified with a nano-sized poly(butyl acrylate) (PBA) layer to improve the hydrophobicity of the GO and improve compatibility with PVDF. The improved hydrophobicity was elucidated using contact angle investigations, and exhibit nearly 0° for neat GO and 102° for GO-PBA. Then, the neat GO and GO-PBA particles were mixed with PVDF using a twin screw laboratory extruder. It was clearly shown that nano-sized PBA layer acts as plasticizer and shifts glass transition temperature from −38.7 °C for neat PVDF to 45.2 °C for PVDF/GO-PBA. Finally, the sensitivity to the vibrations of various frequencies was performed and the piezoelectric constant in the thickness mode, d_33_, was calculated and its electrical load independency were confirmed. Received values of the d_33_ were for neat PVDF 14.7 pC/N, for PVDF/GO 20.6 pC/N and for PVDF/GO-PBA 26.2 pC/N showing significant improvement of the vibration sensing and thus providing very promising systems for structural health monitoring and data harvesting.

## 1. Introduction

Stimuli-responsive systems are of interest to the researcher, due to their smart and in fact reversible behavior upon external input. For this purpose the electric field [1], magnetic field [2], temperature [3], light [4,5,6,7] and pH stimulation [8] are in the majority due to its rise in real-life applications. Another very important smart stimuli-responsive behavior is generation of the electrical output under mechanical excitation also called piezoelectric effect [9]. Such phenomenon finds an enormous application potential, in the field of energy harvesting [10,11,12], tunable mechanical actuation [13,14] and various sensors [15] including monitoring human actions [16,17,18]. In case of sensing applications, the vibration control utilizing such piezo-active systems is very promising in the area of structural health monitoring of civil structures such as bridges [19], they can also be effectively used for aircraft main body structure control [20,21] and finally it seems to also be applicable for controlling of unwanted vibrations of electrical machinery as a source of certain defect exhibiting specific change in vibration profile [22].

For the aforementioned applications, piezoelectric materials such as various ceramics based on lead zirconate titanate (PZT) [23], lead-free ceramics [24], are used in the majority of cases. However, such materials are still very expensive, brittle and the durability of such systems is short [25]. Therefore, the attention is focused on the more flexible materials with permanent piezoelectric properties. The main representative of this group is poly(vinylidene difluoride) (PVDF) [26,27,28]. In this case their copolymers with trifluoro ethylene (TrFE) [29] or hexafluoro propylene (HFP) [30,31] are also frequently used. In order to improve the electro-active properties the addition of some fillers is usually employed, i.e., zinc oxide (ZnO) [32], carbon nanotubes [33], graphene [13,34] or PZT [35]. Some hybrid systems were also successfully applied [36]. Among these fillers, graphene oxide (GO) PVDF composites showed very promising results of piezo-activity [37,38], even at low filler loadings [39]. Generally, the piezo-activity of the PVDF and its copolymers is strongly dependent on the crystalline phase, mainly on the β-phase content. Since the α-phase is non-electroactive [40] the *β* and *γ* provide suitable electro-activity. [41] To obtain the PVDF of desired piezoelectric properties the β-phase has to be developed. For this purpose, there are various processing techniques to obtain such a PVDF system, i.e., stretching [42,43], poling, electrospinning [44,45,46], melt-electrowriting [47] or a combination of them [48,49].

In agreement with the listed literature sources above, the compatibility of the particles and PVDF matrix seems to be crucial. In this respect the modification of the GO particles with various polymers was already developed by our group, when enhanced compatibility was obtained by grafting of poly(methyl methacrylate), poly(butyl methacrylate), poly(styrene), poly(glycidyl methacrylate) and finally, poly(trimethylsilyloxyethyl methacrylate) [50,51,52], and thus their stimuli-responsive capabilities were significantly improved. Moreover, such approach provides a considerably softer and more flexible composite than observed for composites consisting of neat particles [53].

Therefore, our present study is mainly focused on the development of a flexible sensor able to detect the vibrations at a wide range of frequencies and it is still independent of the applied electrical load. Thus, GO grafted particles with poly(butyl acrylate) brushes were synthesized and their effect on the physical as well as vibration sensing capabilities was investigated.

## 2. Materials and Methods

### 2.1. Materials

Graphite (powder, <20 μm,) as a starting material was applied for fabrication of GO particles. Sulfuric acid (H_2_SO_4_, reagent grade, 95–98%), sodium nitrate (NaNO_3_, ACS reagent, ≥99%), potassium permanganate (KMnO_4_, 97%) and hydrogen peroxide (H_2_O_2_, ACS reagent, 29.0–32.0 wt.% H_2_O_2_ basis). Double functional ATRP initiator 2-bromo propionyl bromide (BPB, 98%) was covalently bonded to GO particles. Initiator functionalization was performed with triethyleneamine (TEA, ≥99%). n-butyl methacrylate (n-BA, 99%), ethyl 2-bromopropionate (EBP, 98%), N,N,N′,N″,N″-pentamethyldiethylenetriamine (PMDETA, ≥99%), copper bromide (CuBr, ≥99%) and anisole (ACS reagent, 99%) were also used. Dimethyl formamid (DMF) (ACS reagent, anhydrous, ≥99%) and diethyl ether (ACS reagent, anhydrous, ≥99%) were used as purifying and drying agents. All chemicals were received from (Sigma Aldrich, St. Louis, MO USA) and were used without further purification (except for n-BA). n-BA was purified by passing through a neutral alumina column to remove MEHQ inhibitor prior to its use. Tetrahydrofurane (THF, p.a.), acetone (p.a.), ethanol (absolute anhydrous, p.a.), toluene (p.a.) and hydrochloric acid (HCl, 35%, p.a.) were obtained from (Penta Labs, Prague, Czech Republic). Deionized water (DW) was used during all experimental processes and washing routines. Poly(vinyl fluoride) (PVDF) M_n_ = 107,000 g/mol (Sigma Aldrich, St. Louis, MO, USA) was used as received.

### 2.2. Fabrication of the GO-PBA Hybrid Particles

The GO particles were synthesized and functionalized with ATRP initiator and finally grafted with PBA brushes according to our previously published papers [54,55,56] and is properly described in Figure 1. The last step of modification (polymerization of PBA from the surface of the GO particles) is described as follows. Three grams of GO particles were placed to the 250 mL Schlenk flask and filled with argon. The monomer n-BA (150 mmol), initiator EBP (1.5 mmol), PMDETA (6 mmol) and anisole (30 mL, 50 vol.% of whole mixture) were added under argon atmosphere. Reaction mixture was degassed and 4 freeze-pump-thaw cycles were performed. To the frozen mixture the CuBr was added (1.5 mmol) again under argon flow. The molar ratio between the individual components [n-BA][EPB]:[CuBr]:[PMDETA] was set to [100]:[1]:[1]:[4]. The reaction mixture was put into the pre-heated oil bath and carried out for 16 h at 80 °C. Reaction was controlled over 2 h. The product was filtered using DMF and diethyl ether. Finally, the product was dried using lyophilization upon constant weight and stored in a desiccator.

### 2.3. Characterization of the Neat GO and GO-PBA Particles

^1^H nuclear magnetic resonance (NMR) spectra were recorded at 25 °C using an instrument (400 MHz VNMRS Varian, Tokyo, Japan) with deuterated chloroform (CDCl_3_) as a solvent. The molar mass and polydispersity (Đ) of PBA chains were investigated using gel permeation chromatography (GPC) on the GPC instrument (PL-GPC220, Agilent, Tokyo, Japan) equipped with GPC columns (Waters 515 pump, two PSS SDV 5 μm columns (diameter of 8 mm, length of 300 mm, 500 Å + 105 Å)) and a Waters 410 differential refractive index detector tempered to 30 °C. The samples for NMR spectroscopy and GPC analysis were prepared by dilution with CDCl_3_ and THF, respectively, followed by the purification process, in which they were passed through a neutral alumina column. Transmission electron microscopy (TEM, JEM-2100Plus, Jeol, Tokyo, Japan) was used for investigation of the proper GO and GO-PBA fabrication. The following procedure for sample preparation was used: powders were dispersed in acetone using mechanical agitation for 5 min and 2 min of sonication with subsequent dropping of the dispersion onto a copper grid. Fourier transform infrared (FTIR) spectra (64 scans, resolution of 4 cm^−1^) were recorded on a Nicolet 6700 (Thermo Fisher Scientific, Waltham, MA, USA) within a wavenumber range of 3600–600 cm^−1^, while the ATR technique with a germanium crystal was employed. The spectra were recorded at room temperature. The thermo-oxidation decomposition of the samples was monitored on-line using a thermogravimetric analyzer (TGA) operating in an oxygen atmosphere coupled with FTIR with a help of a Nicolet iS10 equipped with TGA-IR module (Thermo Fisher Scientific, Waltham, MA, USA). The Raman spectra (3 scans, resolution of 2 cm^−1^) were collected on a Nicolet DXR (Thermo Fisher Scientific, Waltham, MA, USA) using an excitation wavelength of 532 nm. The integration time was 30 s, while the laser power on the surface was set to 1 mW. The powders were compressed into the form of pellets (diameter of 13 mm, thickness of 1 mm) on a laboratory hydraulic press (Trystom Olomouc, H-62, Olomouc, Czech Republic). The pellets were used for electrical conductivity measurements as well as for contact angle (CA) determination. The former investigation was performed by two-point method at laboratory temperature with the help of an electrometer (Keithley 6517B, USA). The latter one was evaluated from the static sessile drop method carried out on a surface energy evaluation system equipped with a CCD camera (Advex Instruments, Brno-Komín, Czech Republic). A droplet (5 μL) of PDMS was carefully dripped onto the surface and the CA value was recorded. The presented CA results are the average values from 10 independent measurements.

### 2.4. Composites Fabrication and Poling Procedure

PVDF-based composites containing neat GO as well as GO-PBA particles were mixed using a laboratory twin-screw extruder Brabender (Duisburg, Germany). The temperature was set to 190 °C and 60 rpm of screws was used. Based on our previous research, as the β-phase content in the GO-PVDF composites was significantly enhanced for low filler contents [39], 1 vol. % of particles was utilized. The neat PVDF and GO-based composites were then poled using high electric field (10 kV mm^−1^) by high voltage source (TREK, Advanced Energy, Denver, CO, USA), two electrodes with 25 mm diameter were used to achieve sufficient electric field strength.

### 2.5. Composites Characterization

The crystalline phase, namely, α-phase and β-phase, was investigated in the prepared composites using a MiniFlex600 XRD diffractometer (Japan, RIGAKU) with Co Kα source (operating at 40 kV and 20 mA) and scan range 2 *θ* between 5 and 45°. The calculation of the β-phase content present in the investigated samples was performed using FTIR spectra and Equation (1) was used.
(1)$$F(\beta )=\frac{A_{\beta }}{\frac{\kappa_{\beta }}{\kappa_{\beta }}A_{\alpha }+A_{\beta }}$$
where *A*_α_ and *A*_β_ are values of absorbance corresponding to the wavenumbers 762 cm^−1^ and 840 cm^−1^, respectively. The κ_α_ and κ_β_ are absorption coefficients for α-crystalline phase and β-crystalline phase, having values 6.1 × 10^4^ cm^2^ mol^−1^ and 7.7 × 10^4^ cm^2^ mol^−1^, respectively [47]. The results from Equation (1) are summarized in Table 1. The spectra were collected using a Nicolet 6700 (Thermo Fisher Scientific, Waltham, MA, USA) within a wavenumber range of 3600–600 cm^−1^, ATR mode and germanium crystal. Differential scanning calorimetry (DSC) was used to calculate the crystallinity, *X*_c_, of the GO-based PVDF composites using DSC 1 (Mettler Toledo, Switzerland) in the temperature range from −60 to 230 °C and Equation (2) was employed for this purpose [45]. Results from the DSC investigations are summarized in Table 1:
ijms-23-05777-t001_Table 1Table 1Summarized values of the DSC for neat PVDF and PVDF GO-PBA.Sample code*T*_m_ (°C)Δ*H*_m_ (J g^−1^)*T*_c_ (°C)Δ*H*_c_ (J g^−1^)*X*_c_ (%)neat PVDF171.742.0134.151.740.2PVDF/GO-PBA173.254.6132.862.452.3
(2)$$X_{c}=\frac{\Delta H_{m}}{\Delta H_{m}^{0}}\times 100$$
where Δ*H*_m_ is the heat of fusion for individual samples and Δ*H*_m_^0^ is the heat of fusion obtained for 100% crystalline PVDF (104.5 J g^−1^).

Rheological properties were investigated using a MCR-502 rotational rheometer (Anton Paar, Graz, Austria). The viscoelastic properties were investigated in the linear viscoelastic region at 1% strain deformation. The frequency sweep was set from 0.1 to 10 Hz and temperatures of 150, 170 and 190 °C. Dynamic mechanical analysis at frequency 1 Hz, deformation 0.1% in broad temperature range from −40 up to 100 °C, was investigated using DMA 1 (Mettler Toledo, Greifensee, Switzerland). The dielectric spectroscopy in temperature range from −150 to 100 °C and in frequency range from 10^−1^ to 10^7^ Hz was employed to investigate the polymer chain dynamics using Novocontrol CONCEP 40 (Novocontrol, Montabaur, Germany).

The relaxation process of the side chains was evaluated through activation energies calculated from Arrhenius equation Equation (3) [57]:(3)$$f_{\beta }=f_{\infty }\exp\left(\frac{E_{a}}{k_{B}T}\right)$$
where *E*_a_ is the activation energy and *f*_∞_ is the pre-exponential factor and *T* is temperature in Kelvin and *k*_B_ is Boltzmann constant.

Other relaxation processes connected to the main backbone movement where calculated using the Vogel–Fulcher–Tamman equation Equation (4) [58]:(4)$$f=f_{0}\exp\left(\frac{E_{a}}{k_{B}\left(T-T_{0}\right)}\right)$$
where, *f* is the relaxation frequency, *f*_0_ is the pre-exponential factor, *E*_a_ is the activation energy, *T* is the thermodynamic temperature, *T*_0_ is Vogel temperature and *k*_B_ is the Boltzmann’s constant.

Vibration sensing capability of the fabricated composite systems based on neat GO and GO-PBA particles mixed with PVDF matrix was measured in the thickness mode and d_33_ piezoelectric constant was performed similarly as in our previous publications [59,60].

## 3. Results and Discussion

### 3.1. Grafting of the Neat GO Particles with Poly(Butyl Acrylate) Brushes

The successful polymerization procedure was confirmed using ^1^HNMR from which the presence of the polymer can be obvious (Figure 2a) and the corresponding amount of unreacted monomer was also calculated. Then the PBA polymer was analyzed using GPC and molecular weight and polydispersity index was elucidated (Figure 2b). It was confirmed that, resulting Mn of the final PBA was 5400 g mol^−1^, which is in good agreement with 44% calculated from ^1^HNMR spectra. Very narrow PDI, clearly indicated that synthesized polymer is in the form of brushes. All results are summarized in the Table 2.

**Table 2 ijms-23-05777-t002:** Reaction conditions for the performed grafting of PBA from the GO surface.

Sample name	M ^a^	I ^a^	L ^a^	CuBr	*M*_n_^b^ (g mol^−1^)	*Đ* ^b^	Conversion ^c^ (%)
GO-PBA—2 h	100	1	4	1	5400	1.16	44

^a^ M, I, L represents the monomer, sacrificial initiator and ligand; ^b^ according to GPC; ^c^ according to ^1^H NMR.

The successful fabrication of the neat GO from the graphite using modified Hummers method is confirmed in the Figure 3a. The well-exfoliated GO can be clearly seen as only a single sheet is present, similarly as that also shown by other research groups [61]. The modification of the GO by PBA brushes grafted from the surface is visible as a flossy-like coverage in Figure 3b and showing the successful coating.

As another confirmation of the successfully synthesized GO and GO-PBA hybrid sheets, TGA with online monitoring of FTIR during decomposition was performed and evaluated (Figure 4). It can be seen that neat GO contains oxygen containing groups whose decomposition temperatures start from 180 °C and finish at 260 °C, which was also shown as absorption bands from FTIR for -OH (3510 cm^−1^), C=O (1723 cm^−1^) and C-OH (1428 cm^−1^). The coating of the GO with PBA brushes can be clearly seen in the range from 240 to 420 °C, which is the typical range for decomposition of the PBA material [62]. In this range the FTIR signal was also collected and provided information about the present groups corresponding mainly to the PBA coating. Absorption bands for CH_3_ and CH_2_ are presented at 2965 cm^−1^ and 2742 cm^−1^, in the cases of C=O and C-O-C the signal at 1728 cm^−1^ and 1428 cm^−1^ was observed, respectively.

Since sustainable morphology is very important for the application of the GO sheets as well as their non-conducting nature, the Raman spectra of both powder systems were performed. It can be seen, that in both cases, the 2D structure of the graphene was observed as the absorption in the range from 2500 to 3000 cm^−1^. Thus, the modification of GO with PBA polymer brushes does not significantly affect the morphology, which is in good agreement with the TEM observations. Moreover, the conductivity of the GO systems is very important from the vibration sensing applicability point of view. PVDF based composites exhibiting conducting character are not capable of being efficient as electromechanical actuators or sensors to collect sufficient signal from vibration. In our case, the GO-PBA powder has conductivity only 9.6 × 10^−6^ S cm^−1^ indicating that if we use only 0.1 vol. % of powder in PVDF, we are still under percolation and therefore not providing the conducting pathway, only just increasing the amount of the dipoles in the system, which should positively contribute to the enhanced vibration sensing capability. However, as was already observed in our group, the conductivity of the GO can be controllably tuned using surface-initiated atom transfer radical polymerization (SI-ATRP) procedure in various manners. For very fine tuning, to achieve the system with slightly higher conductivity than GO, low ratio between catalyst and ligand and temperatures below 80 °C have to be utilized [51]. On the other hand, when the conductivity needs to be decreased significantly and the polymer shell needs to be well-developed, the temperature of 80 °C and high ratio between the ligand and catalyst needs to be used [56]. The reduction of the GO can be confirmed by the Raman spectra when absorption bands for D and G peaks highlighted in Figure 5 are evaluated. In this case, the ratio between the I_D_ to I_G_ is usually less than one for non-conducting and oxidized forms of graphene oxide and higher than one for reduced forms. In our case, only a slight increase was observed from 0.92 to 1.05 for neat GO and GO-PBA, respectively.

Compatibility of the particles and polymer matrix plays a crucial role for the overall performance of the prepared composites mainly from the mechanical point of view. In this case, since the GO surface is strongly hydrophilic, the water drop was not able to be measured. The drop which is present in Figure 6a is just to represent the place where the drop was deposited. On the other hand, after the modification of the GO with PBA brushes, the surfaces of the particles change rapidly and calculated contact angle was 102° (Figure 6b), showing the considerably changed surface properties and thus the proper compatibility of the filler with PVDF can be expected and was investigated further using rheological measurements.

For the investigation of the compatibility between the GO particles and PVDF matrix, rheological investigation was performed (Figure 7). In the previous study, we observed significantly improved compatibility for GO modified with poly(methyl methacrylate) in elastomeric matrix in comparison to neat GO by shifting of the crossover points to higher frequencies [63] which was also observed by other research groups. In our case, the crossover point is not visible, however, the trends at various temperatures are clearly observed and show that possible crossover points will appear for GO-PBA-based composite at the latest. Generally, the neat GO-based composites behave very similarly to that of the neat PVDF matrix, however, significant softening, due to the presence of short polymer brushes, was observed for PVDF/GO-PBA composites.

Changes in the compatibility between the GO-PBA and PVDF are reflected in the primary crystalline structure. Especially, for the composites containing the PVDF, this crystallinity is very crucial from the applicability point of view. Therefore, the transformation of the α-phase to β-phase was calculated. In the case of all samples (Figure 8) the peaks for these crystalline phases are assigned. The neat PVDF showed relatively high amount of the α-phase since the peaks at 762 cm^−1^ and 795 cm^−1^ are clearly visible. On the other hand, the neat GO added to the PVDF causes considerable enhancement of the β-phase as was already published [34,39] and the α-phase was suppressed as well. However, the modification of the GO with PBA brushes provides more restricted development of the α-phase and on the other hand, even enhanced β-phase in comparison to PVDF/GO composite. In this case, the β-phase was calculated to be 39%, which is slightly lower than published by others [45,47], however, still very close. However, the PVDF composites showing considerable increase to 76% and 85% for neat PVDF-GO and PVDF/GO-PBA, respectively.

In order to confirm that the transformation of the crystalline α-phase to β-phase was successfully performed, the XRD as a comparative method to FTIR was used and presented as is usual for similar PVDF-based systems [44,45]. As can be seen in the Figure 9, in the neat PVDF sample the peaks for α-phase are present at 20.8° and 21.6° while for the β-phase they are present at 23.4°. However, after the addition of the neat GO particles into the PVDF, the peaks for α-phase were considerably suppressed and peak for β-phase was slightly shifted to the higher angles. Moreover, such phenomenon was even enhanced for composites containing GO-PBA particles and the β-phase peak appeared as more significant, due to the presence of the GO-PBA particles as an active filler. The values of the peaks for β-phase were found to be 23.7° and 24.1 for PVDF/GO and PVDF/GO-PBA, respectively.

#### 3.1.1. Differential Scanning Calorimetry

To confirm the improvement in the development of the crystallinity which is a crucial factor influencing the d_33_ coefficient, the calorimetric properties (Table 2) of the prepared samples were investigated. As can be seen the amount of the crystalline phase was enhanced, due to the presence of the GO-PBA particles, these act as nucleating agents. The melting enthalpy, Δ*H*_m_, increased from 42.0 to 54.6 J g^−1^. Moreover, the enthalpy of crystallization increased as well. Final crystallinity, *X*_c_, increased significantly from 40.2% for neat PVDF to 52.2% for PVDF/GO-PBA. Therefore, also as expected, that higher amount of the β-phase will finally be transformed as was confirmed by XRD and FTIR investigations.

#### 3.1.2. Dynamic Mechanical Analysis

The mechanical properties are very important for the application of the PVDF as structural health monitoring sensors, since those sensors are exposed to significant vibrations, and thus dynamic mechanical stimulation. In this respect, the DMA investigation was performed and evaluated in a broad temperature range from −150 to 100 °C (Figure 10). This range was chosen, due to the possible investigation of the T_g_ and impact of the grafting on the dynamic mechanical capability. It is not expected that PVDF-based sensors will be applied below −60 °C and above 80 °C; even though the storage modulus is very stable up to −145 °C. Generally, the storage modulus is very similar for all investigated samples. Below the T_g_, the highest values were obtained for sample PVDF/GO-PBA, and this trend continued above and from approximately 25 °C, the storage moduli (Figure 10a) starts to drop but in a similar way for all investigated samples. In case of tan *δ* investigations, the position of the T_g_ was mainly estimated. Neat PVDF exhibits T_g_ at −38.7 °C, which is in agreement with other literature sources [64]. The slight shift to lower temperatures was also obtained for sample PVDF/GO, however, significant shift of lower temperatures to −45.2 °C (Figure 10b) is visible in sample PVDF/GO-PBA, due to the presence of the short polymer brushes, those acting as a plasticizer. Similar behavior was also already published by our group in the case of carbonyl iron [53] as well as carbon nanotubes [65]. Furthermore, the value of the tan *δ* is significantly higher, showing that samples are more ductile in the whole investigated temperature range. Finally, it can be seen that sample PVDF/GO-PBA exhibits more plasticized structure, and thus can provide better sensing capability upon vibration stimulation as is shown in the last part of this paper.

#### 3.1.3. Dielectric Properties

Broadband dielectric spectroscopy was used as a useful tool for characterization of the polymer chain dynamic for neat PVDF as well as composites containing GO and GO-PBA particles. For all investigated systems the main three relaxations were found (Figure 11a–c), similarly as was published elsewhere [66] in the case of PVDF (*α* and *β* relaxations) and also that such kinds of semicrystalline polymers exhibit interfacial, also called Maxwell–Wagner–Sillars (MWS) [67]. The β relaxation is connected to the side fluorine groups and can be visible in the temperature range −90 up to −60 °C. The relaxation as a main polymer chain relaxation is visible close to −50–0 °C depends on the applied temperature. Finally, MWS relaxation, which is very usual for PVDF-based materials and composites, are present at high temperatures and low frequencies [68,69]. However, to be able to clearly see this MWS relaxation, the relative permittivity and dielectric losses were recalculated to the dielectric loss modulus expression similarly as in our previous publications [70,71]. As can be seen from Table 3, the activation energy calculated from the dielectric map (Figure 11d), β relaxation increases with addition of the neat GO particles and reaches the highest values 25.9 kJ mol^−1^ for PVDF/GO-PBA, indicating that the motion of the fluorine side groups are significantly restricted and fixed in the β-phase position as was confirmed by XRD and FTIR. In the case of the main polymer chain, the activation energy decreases with GO addition indicating a more flexible polymer chain and is the lowest (6.0 kJ mol^−1^) for the PVDF/GO-PBA sample, showing that T_g_ was significantly decreased, due to the presence of the short PBA chains, those acting as plasticizers. These results are in agreement with those obtained in the case of results from dynamic mechanical analysis. Finally, the activation energy calculated for MWS relaxation is the highest for the neat PVDF, most probably due to the presence of the α-crystalline phase. Further, development and suppression of the α-phase substantially decrease the value of activation energy and allow the ion more intensified transport and reaction of the electric field since the relaxation time is the lowest for sample PVDF/GO-PBA.

#### 3.1.4. Vibration Sensing

In the framework of all the performed experiments and characterizations in this paper and to show that SI-ATRP modification of the GO is a useful tool for the implementation of this techniques for vibration sensor fabrication, vibration sensing capabilities were investigated. It can be seen that neat PVDF showing the lowest response to the vibration (Figure 12a) in comparison to the PVDG/GO (Figure 12b) and PVDF/GO-PBA (Figure 12c). However, the response can be intensified by increased used resistance. Furthermore, the response on the samples is smaller for low applied frequencies of vibration and further increases. This phenomenon is similar to those obtained is our previous work [60,72]. Finally, the d_33_ piezoelectric constant was calculated and it can be seen (Figure 13) that neat PVDF shows 14.7 ± 0.1 pC/N. The presence of the neat GO as the filler slightly improves it up to 20.6 ± 0.4 pC/N, due to the possible larger amount of charges present on the interface of GO. Finally, the PVDF/GO-PBA shows very promising 26.2 ± 0.1 pC/N, mainly due to the proper polymer dispersion due to the enhanced contact angle, better charge transport and also electromechanical properties due to the well-developed β-crystalline phase. All these benefits were possible due to the application of the short PBA brushes on the surface of GO particles.

## 4. Conclusions

In this paper, the GO-PBA hybrid particles were synthesized using the SI-ATRP approach and provide nanometer scale coating on the surface of GO. The nature of polymer and basic characterizations were performed using NMR and GPC techniques. The confirmation of the presence of the PBA coating on the GO particles was done using a TGA-FTIR coupled device and TEM microscopy. The main changes in the electrical conductivity after coating with PBA was investigated using a four-probe method and Raman spectroscopy. The significant change of the surface properties and thus improved compatibility between GO-PBA particles and PVDF was due to the significantly enhanced water contact angle from nearly 0° for GO and 102° for GO-PBA particles. The fabricated composites of GO and GO-PBA and PVDF using a twin-screw extrusion process were analyzed from the structural and physical properties point of view. It was investigated that due to the presence of the GO-PBA containing short polymer brushes and more restricted situation in PVDF, the β-phase transformation was more successful (85%) in comparison to neat PVDF (39%) and PVDF/GO (76%). This observation was also confirmed by XRD and moreover DSC results showed increased crystallinity from 40.2% for neat PVDF to 52.3% for PVDF/GO-PBA. The one main finding in this paper is that nano-sized PBA layers act as plasticizers in the case of PVDF matrix and shift the glass transition temperature from −38.7 to −45.2 °C for neat PVDF and PVDF/GO-PBA, respectively. Similar findings were confirmed by dielectric spectroscopy, where the activation energy of α relaxation for PVDF/GO-PBA is the lowest and reaches 6.0 kJ mol^−1^. Finally, the vibration sensing in the broad frequency range and various load resistances were investigated and it was shown that due to the proper GO-PBA dispersion, a more flexible composite based on GO-PBA was prepared; the highest d_33_ piezoelectric constant (26.2 pC/N) in comparison to neat PVDF (14.7 pC/N) and PVDF/GO (20.6 pC/N) was obtained.

## Figures and Tables

**Figure 1 ijms-23-05777-f001:**
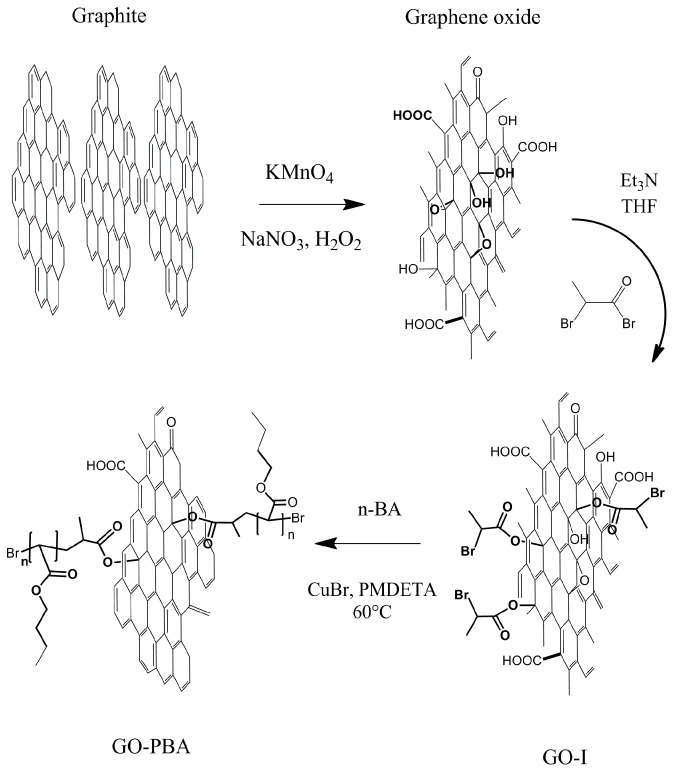
Schematic illustration of the GO modification with PBA polymer brushes.

**Figure 2 ijms-23-05777-f002:**
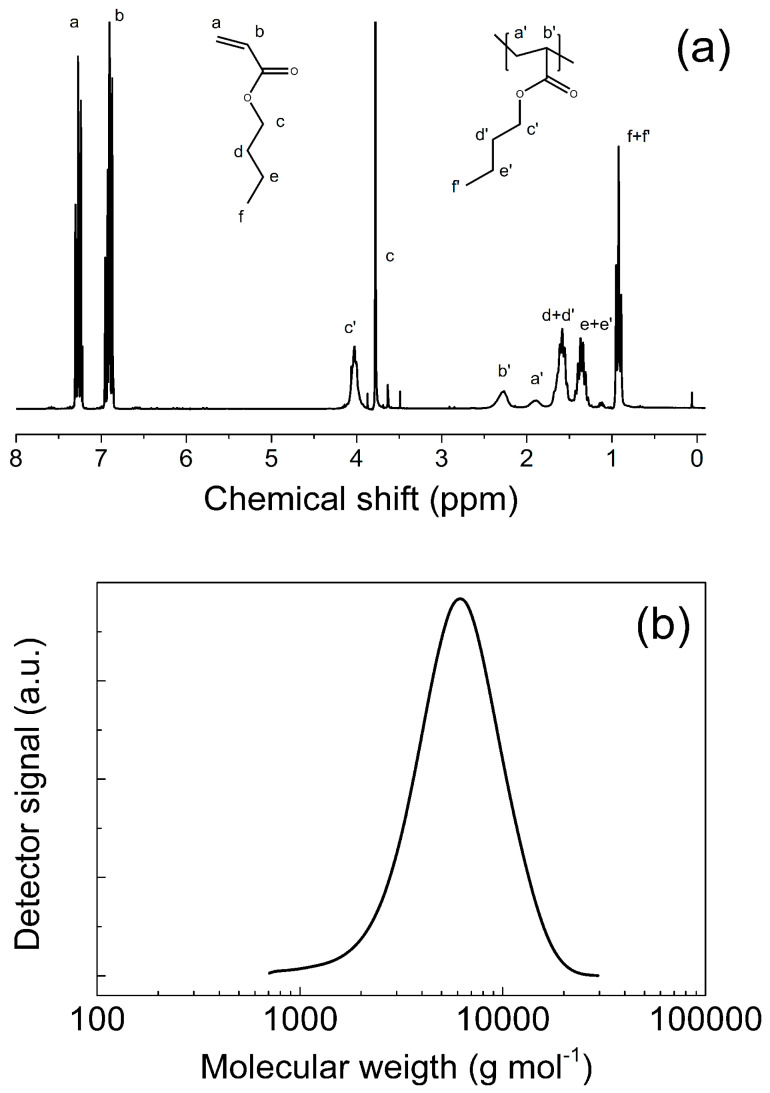
^1^HNMR spectra for n-butyl acrylate and PBA (**a**) and GPC trace of the controlled polymerization of PBA (**b**).

**Figure 3 ijms-23-05777-f003:**
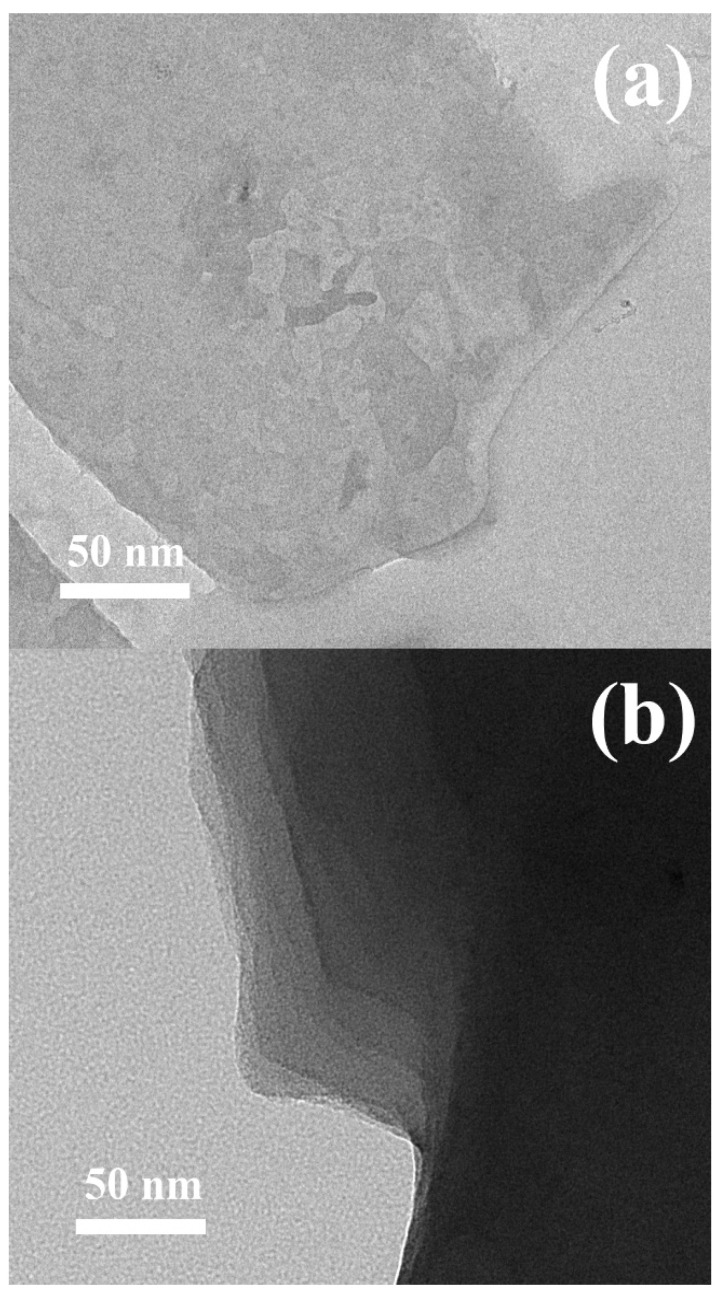
TEM images of the neat GO (**a**) and GO-PBA hybrid particle (**b**).

**Figure 4 ijms-23-05777-f004:**
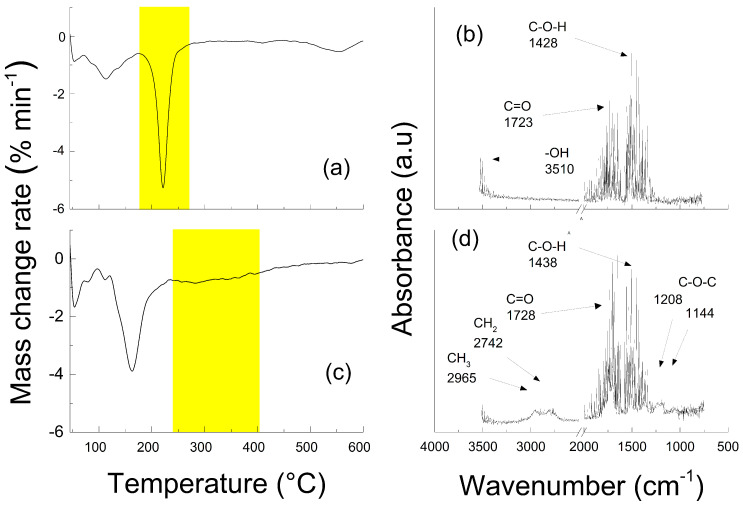
TGA spectra of neat GO (**a**) and GO-PBA hybrid particle (**b**) and corresponding FTIR evaluation during sample decomposition of neat GO (**c**) and GO-PBA (**d**). Yellow region representing the stage, when the FTIR signal was collected.

**Figure 5 ijms-23-05777-f005:**
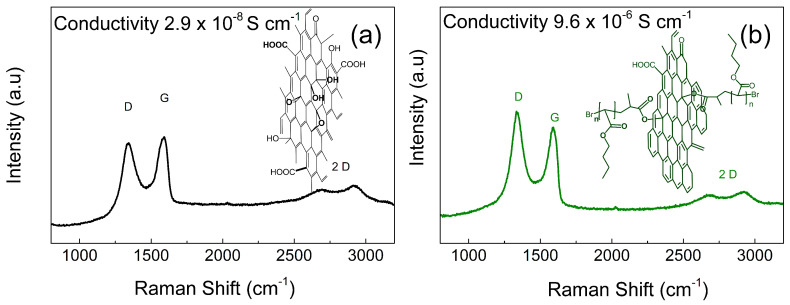
Raman spectra and electrical conductivities for neat GO (**a**) and GO-PBA (**b**) particles.

**Figure 6 ijms-23-05777-f006:**
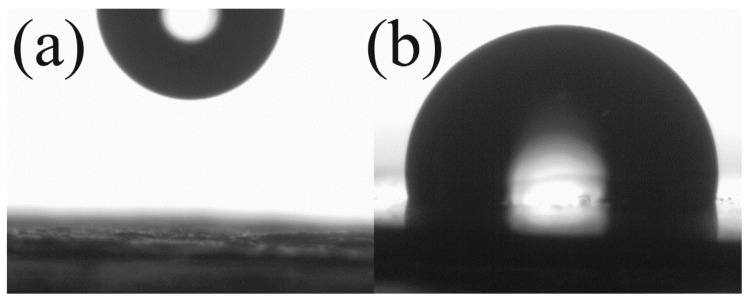
Contact angle measurements using sessile drop method of neat GO (**a**) and GO-PBA (**b**).

**Figure 7 ijms-23-05777-f007:**
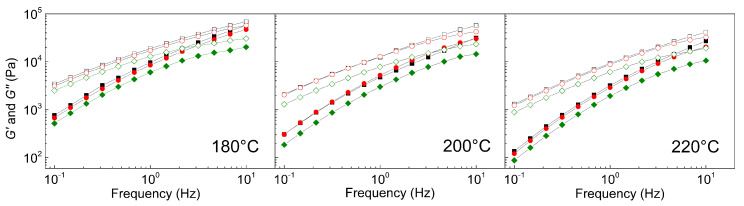
Dependence of the storage (solid symbols) and loss (open symbols) moduli against frequency, for neat PVDF (black squares), PVDF/GO (red spheres) and PVDF/GO-PBA (green diamonds) at various temperatures.

**Figure 8 ijms-23-05777-f008:**
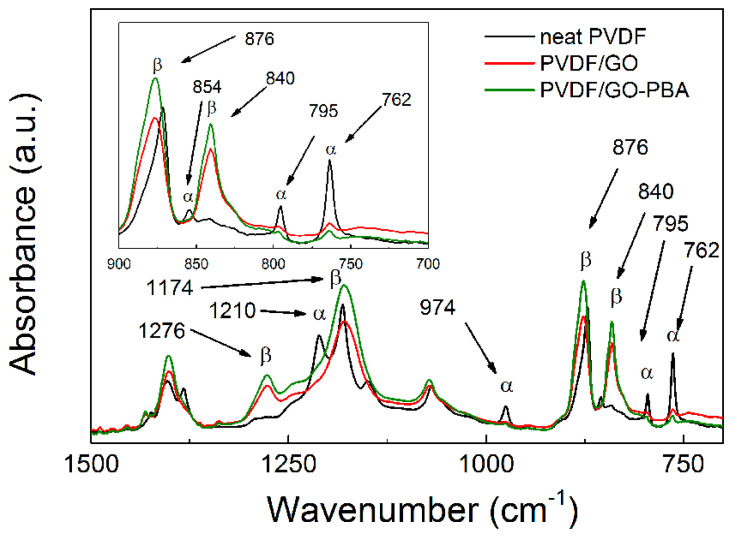
FTIR spectra of the neat PVDF and composites containing PVDF/GO and PVDF/GO-PBA particles.

**Figure 9 ijms-23-05777-f009:**
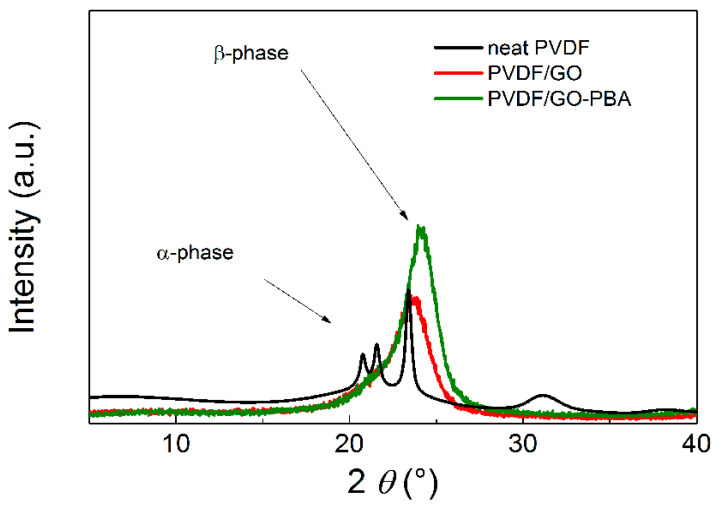
XRD spectra of the neat PVDF and composites containing PVDF/GO and PVDF/GO-PBA particles.

**Figure 10 ijms-23-05777-f010:**
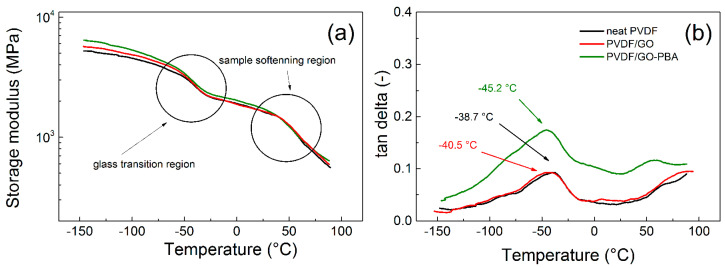
Temperature dependence of storage modulus (**a**) and tan delta (**b**) for neat PVDF and composites containing PVDF-GO and PVDF-GO-PBA particles.

**Figure 11 ijms-23-05777-f011:**
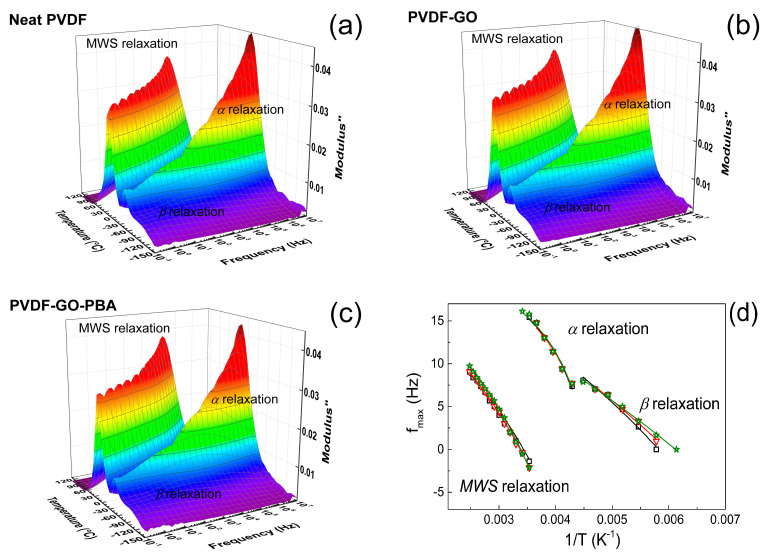
Dielectric 3D plots of neat PVDF (black squares) (**a**) and composites containing PVDF/GO (red down triangles) (**b**) and PVDF/GO-PBA particles (green stars (**c**). Dielectric map of the relaxations present in the PVDF-based systems (**d**). Solid lines in the dielectric map are the best fit of the Arrhenius and VFT model.

**Figure 12 ijms-23-05777-f012:**
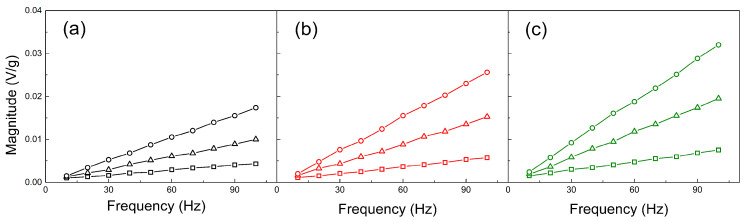
Vibration sensing measurements of neat PVDF (black) (**a**) and composites containing PVDF/GO (red) (**b**) and PVDF/GO-PBA particles (green) (**c**) at various frequencies and resistances—10 kΩ (squares), 30 kΩ (triangles) and 50 kΩ (circles).

**Figure 13 ijms-23-05777-f013:**
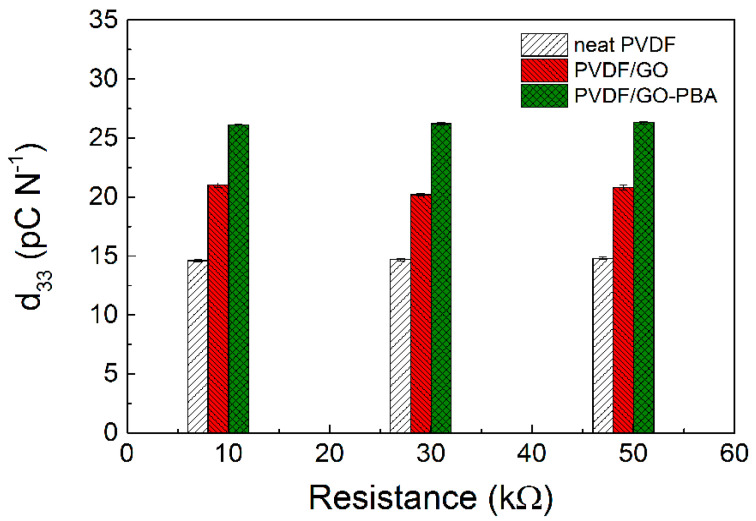
Identification of the d_33_ piezoelectric constant at various resistances for neat PVDF, PVDF/GO and PVDF/GO-PBA.

**Table 3 ijms-23-05777-t003:** Summarized parameters of the Arrhenius and VFT model fit for fabricated samples.

Sample Name	VFT α-Relaxation PVDF			VFT MWS-Relaxation PVDF			Arrhenius β-Relaxation -CF_2_
	τ_0_ (s^−1^)	T_0_ (K)	B (kJ mol^−1^)	τ_o_ (s^−1^)	T_0_ (K)	B (kJ mol^−1^)	E_a_ (kJ mol^−1^)
Neat PVDF	2.1·10^−11^	187	6.5	2.84·10^−11^	125	31.0	21.0
PVDF/GO	2.1·10^−11^	185	6.4	2.66·10^−11^	136	29.9	22.2
PVDF/GO-PBA	2.1·10^−11^	182	6.0	2.53·10^−11^	153	26.7	25.9

## Data Availability

Not applicable.

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
