# Peer review of "Effect of Nano-Sized Poly(Butyl Acrylate) Layer Grafted from Graphene Oxide Sheets on the Compatibility and Beta-Phase Development of Poly(Vinylidene Fluoride) and Their Vibration Sensing Performance"

_ijms, 2022, doi:10.3390/ijms23105777_

Round 1
Reviewer 1 Report
This manuscript reports the modification of graphene oxide (GO) particles with poly(butyl acrylate) to improve the hydrophobicity of the GO and the compatibility with PVDF. The authors proved the enhancement in hydrophobicity using contact angle and the compatibility with PVDF using shift in glass transition temperature. The authors also report the application to piezoelectric sensor for vibration sensing. The investigation and conclusion described in this study are sound and promising toward sensor industry. However, I have some issues with the manuscript as written that must be addressed before it is suitable for publication in International Journal of Molecular Sciences. I recommend a publication after minor revision with additional data and a better structuring of the findings.
- Change of notation is recommended to distinguish difference between blend and chemical bonding. In other words, GO modified with PBA can be “GO-PBA”, but blending PVDF with GO-PBA should be “PVDF/GO-PBA”.
- Misspelled manuscript (page2 line 53, and line 54, page 11, line 319) : a should be alpha and b should be beta.
- Page 3, line 109 : “PMMA chains”, where these PMMA suddenly come from? Misspelling of PBA?
- Page 3, line 111 : “two PPS SDV 5 mm columns”, isn’t it be “PSS” column?
- Page 14, line 384 : “PVDG-GO” should be PVDF-GO
- In order to know the applicability as a vibration sensor, it is important to maintain the compatibility of the polymer blend (PVDG/GO-PBA), so it is necessary to test it (i.e. creep test and AFM morphology under and after application as a sensor).
Author Response
The notes to reviewers are attached.

Reviewer 2 Report
In their manuscript “Effect of Nano-Sized PBA Layer Grafted from Graphene Oxide Sheets on the Compatibility and Beta-Phase Development of PVDF and Their Vibration Sensing Performance” the authors fabricated a PVDF composite with graphene oxide(GO)-PBA filler with improved hydrophobicity and piezoelectric properties. The composite improved the piezoelectric coefficient d_33 by a factor of almost 2 from that of neat (pure) PVDF.
Their investigations are thoroughly in order to characterize the fabricated materials.
There are a few questions their results might clarify. As I understand the enhancement of piezoelectric properties is mainly due to the thermal treatment in the presence of GO/PBA by which PVDF undergoes a transition from alpha phase to beta phase. Is any estimate of how much of alpha phase is transformed into beta phase? A quick estimate would be the factor by which d_33 went up. Please comment on this issue!
Another question regards dielectric measurements. By temperature-dependent dielectric measurements the authors identified 3 relaxation mechanisms common in polymers: alpha and beta as well as Maxwell-Wagner-Sillars structural relaxation. Would the authors explain how they identified and discriminated these three dielectric relaxation mechanisms? For instance, for Maxwell-Wagner-Sillars mechanism the relaxation time is roughly related to the ratio between the real part of dielectric permittivity of the components and the DC conductivity. Thus a temperature-dependent measurement of DC conductivity might reveal the conduction mechanism with temperature-dependent relaxation times similar to those of the Maxwell-Wagner-Sillars mechanism identified in the text.
The authors should proofread the manuscript more carefully. There is no equation (5) on page 4, etc..
Author Response
The notes to reviewers are attached.

Reviewer 3 Report
The article involves the study of PVDF-GO nanocomposites for sensor development. In my opinion, this manuscript is interesting and well-written. Data are well presented and discussed. I recommend this manuscript for publication, but there are some typo mistakes throughout the manuscript that need to be fixed: the name of the crystalline phases of PVDF must have Greek letters, mistakes in writing the words in English and some figures need to be revised - the graphs are 1/3 of the picture, losing resolution (at fig. 4, 5, 6, 9, 10, 12) and for fig 8 the medallion is useless.
Author Response
The notes to reviewers are attached.
